# Adsorption of 2,4-D and MCPA Herbicides on Carbon Black Modified with Hydrogen Peroxide and Aminopropyltriethoxysilane

**DOI:** 10.3390/ma15238433

**Published:** 2022-11-26

**Authors:** Izabella Legocka, Krzysztof Kuśmierek, Andrzej Świątkowski, Ewa Wierzbicka

**Affiliations:** 1Department of Polymer Technology and Processing, Łukasiewicz-Industrial Chemistry Institute, 01-793 Warsaw, Poland; 2Institute of Chemistry, Military University of Technology, 00-908 Warsaw, Poland

**Keywords:** 2,4-D, MCPA, adsorption, carbon black, surface modification

## Abstract

The carbon black N-220 surface was subjected to modification through H_2_O_2_ oxidation and deposition of aminopropyltriethoxysilane. The pristine (CB-NM) and modified materials (CB-Ox and CB-APTES) were characterized by N_2_ adsorption–desorption isotherms, scanning electron microscopy, energy-dispersive X-ray spectroscopy (SEM-EDS), thermogravimetry, and FTIR spectroscopy. Carbon black samples were applied as adsorbents for the removal of 2,4-dichlorophenoxyacetic acid (2,4-D) and 2-methyl-4-chlorophenoxyacetic acid (MCPA) herbicides from aqueous solutions. The influence of their surface properties on adsorption efficiency was analyzed and discussed. The results showed that the adsorption of the herbicides was pH-dependent, and the most favorable adsorption was observed in an acidic environment. The experimental data best fit pseudo-second-order and Langmuir models for kinetic and equilibrium data, respectively. The adsorption rate of both the herbicides increased in the order of CB-APTES < CB-Ox < CB-NM and was closely correlated with the mesopore volume of the carbon blacks. The monolayer adsorption capacities were found to be 0.138, 0.340, and 0.124 mmol/g for the adsorption of 2,4-D and 0.181, 0.348, and 0.139 mmol/g for the adsorption of MCPA on CB-NM, CB-APTES, and CB-Ox, respectively. The results showed that the surface chemistry of the adsorbent plays a more important role than its porous structure. Both herbicides were preferably adsorbed on APTES-modified carbon black and were adsorbed the worst on oxidized carbon black (CB-APTES > CB-NM > CB-Ox).

## 1. Introduction

Carbon black is a commercial form of solid carbon that is produced by precisely controlled processes to produce custom-designed aggregates of carbon particles that differ in particle size and shape, porosity, aggregate size, and chemical surface properties [1].

Carbon black consists mainly of pure carbon, which is usually 98% by weight, and very small additions of oxygen, hydrogen, and nitrogen. This number may vary based on its production process and final desired application. Carbon blacks are formed by the strictly controlled thermal decomposition of carbon-rich raw materials under oxygen-poor conditions (partial combustion) or in an oxygen-free atmosphere (pyrolysis). The manufacturing process creates carbon black particles with sizes ranging from 10 nm to about 500 nm. These particles are assembled into chain-like aggregates that define the structure of the various types of carbon black. 

The presently known furnace, thermal, channel, lamp, gas, and acetylene blacks are all different types of carbon blacks, each produced using different processes, conditions, and/or starting materials [1,2]. 

Carbon black is the most widely used and cost-effective rubber-reinforcing agent in tire components and other rubber goods [3,4]. The reinforcing potential of carbon black is closely related to its surface activity dependent on the primary particle size and primary aggregate structure [5]. Carbon black is mainly used to reinforce rubber but also as an additive to enhance material performance, including viscosity, conductivity, or UV resistance. They are used quite widely in the polymer, coating, and printing industries (including as a black pigment) as well as various other special applications. Recent scientific results suggest other applications of carbon black that are currently of interest such as renewable energy devices, electrochemical energy storage, and environmental remediation. Others can be listed here, for example, fuel cells, batteries, supercapacitors, photocatalysts, solar devices, carbon dioxide storage, and separations [6]. 

The objective of the present study is to investigate the effectiveness of carbon blacks with different surface chemical properties as adsorbents of chlorophenoxy herbicides (2,4-dichlorophenoxyacetic acid and 4-chloro-2-metylphenoxyacetic acid) from aqueous solutions. These adsorbates belong to halogenated organic contaminations of surface water, groundwater, and drinking water. The 2,4-dichlorophenoxyacetic acid (2,4-D) and its analog 4-chloro-2-metylphenoxyacetic acid (MCPA) are commonly used as herbicides, but as a result of their widespread production and use, are present in water and soil. Both of these chemicals are synthetic auxins. They act similarly to indole-3-acetic acid, which occurs naturally in plants and is a hormone responsible for regulating plant growth and development [7,8]. Their presence in the environment is undesirable, especially since both compounds are highly toxic to living organisms [9]. Therefore, the priority issue becomes the problem of their degradation and prevention of their entry into the ecosystem.

Among methods currently used to remove 2,4-D and MCPA from aqueous solutions, adsorption is one of the well-established and effective techniques. The efficiency of the adsorption process depends on the physical properties of the adsorbent used, but also on the chemical properties of the adsorbent surface. The effect of porous structure on adsorption has already been fairly well-studied and is well-documented (in general, a higher specific surface area of an adsorbent translates into its better adsorption capacity and greater efficiency in removing contaminants from water). Unfortunately, the literature on the effect of adsorbent surface chemistry on the adsorption of phenoxyacetic herbicides is inadequate, and this topic requires further research. Zhu et al. [10] studied the effect of surface modification of porous biochar on the adsorption of 2,4-D. Biochar was modified by surface oxidation and surface amination. The results revealed that specific surface area was not the only factor determining adsorption capacity, and that surface modification significantly affects the adsorption process. The authors concluded that nitrogen-containing functional groups can cause electrostatic repulsion, which slows down the adsorption rate. They also found that surface oxidation destroys the pore structure, but at the same time, oxygen-containing functional groups increase 2,4-D adsorption. The effect of the surface functional group (i.e., 3-aminopropyltriethoxy-, 3-mercaptopropyl-, and n-octyldimethyl-) on adsorption of 2,4-D on hexagonal mesoporous silicate was reported by Patiparn and Takizawa [11]. The results showed that 2,4-D adsorbed better and easier on the adsorbent with positively charged amine groups due to electrostatic interaction. A similar phenomenon (improved adsorption of phenoxyacetic herbicides) was also observed on other APTES-modified materials including mesoporous silica [12,13], mesoporous carbons [14], and a magnetic nanoadsorbent (Fe_3_O_4_@SiO_2_) [15]. Our previously published work [16] investigated the adsorption of three phenoxy herbicides including MCPA, 2-(4-chloro-2-methylphenoxy) propanoic acid, and 4-(4-chloro-2-methylphenoxy) butanoic acid on the activated carbons with different surface chemical properties. As adsorbents, nonmodified Norit R3-ex activated carbon, as well as activated carbon oxidized with nitric acid (acidic surface) and activated carbon heat-treated in ammonia at 900 °C (basic surface), were used. The results showed that the availability of the functional groups on the adsorbent surface was more important than the surface area. All herbicides were most preferably adsorbed on the activated carbon with a basic surface, followed by the nonmodified adsorbent, and the worst adsorption occurred on the activated carbon with an acidic surface. 

In this paper, the adsorption of 2,4-D and MCPA on carbon blacks with different properties—origin carbon black (CB-NM) and carbon blacks modified with hydrogen peroxide (CB-Ox) and aminopropyltriethoxysilane (CB-APTES)—was investigated. To the best of our knowledge, no study has been reported on the effect of carbon black surface chemistry on the adsorption of phenoxy herbicides from aqueous solutions. Adsorption kinetics, adsorption under equilibrium conditions, and the effect of solution pH on the adsorption process were examined.

## 2. Materials and Methods

### 2.1. Reagents and Materials

The 2,4-dichlorophenoxyacetic acid (2,4-D), 2-methyl-4-chlorophenoxyacetic acid (MCPA), and aminopropyltriethoxysilane (APTES, C98%) were purchased from Sigma-Aldrich (St. Louis, MO, USA). The physicochemical properties of the herbicides are shown in Table 1. All other analytical-grade reagents and chemicals were purchased from Avantor Performance Materials (Gliwice, Poland). The carbon black Corax N-220 used in our work was from Orion Engineered Carbons.

### 2.2. Adsorbents Preparation and Characterization

The used carbon black Corax N-220 was divided into three samples: original nonmodified (CB-NM), oxidized with 30% H_2_O_2_ (CB-Ox), and modified by deposition of aminopropyltriethoxysilane on its surface (CB-APTES). The process was carried out in an ethanol environment (1.5 g APTES and 15 g carbon black). After mixing with a stirrer at room temperature for 3 h, the solvent was evaporated, and modified carbon black was obtained.

For characterizing the porous structure of the original and modified carbon black samples, the low-temperature N_2_ adsorption–desorption isotherms were measured using a TriStar II 3020 V1.03 apparatus (Micromeritcs Company, Norcross, GA, USA). On their basis, the Brunauer–Emmett–Teller (BET) surface areas and pore volumes were calculated. Characterization of their surface morphology as well as the chemical composition was carried out with the use of scanning electron microscopy with an energy-dispersive X-ray spectrometer (JSM–6490LV series, JEOL Company, Peabody, MA, USA). Surface chemistry of carbon black samples was also investigated by thermogravimetric analysis performed with a TA Instruments Q50 Thermogravimetric Analyzer in a nitrogen atmosphere of 50 mL/min in the temperature program from ambient temperature up to 700 °C at a heating rate of 10 °C/min. For carbon black samples’ surface chemistry characterizing, the FTIR spectra were additionally recorded with the use of Nicolet iS10 FTIR Spectrometer (Thermo Scientific, Waltham, MA, USA). The analysis was carried out at room temperature in the range of 4000 cm^−1^ to 500 cm^−1^ using KBr tablets (mass ratio 1:240). The point of zero charge (pH_PZC_) was determined by the method reported elsewhere [17].

### 2.3. Adsorption from Aqueous Solutions

All adsorption experiments were carried out in Erlenmeyer flasks into which 10 mL of 2,4-D or MCPA solutions of appropriate concentrations and 0.02 g of adsorbent were introduced. The mixtures were shaken at 25 °C at 200 rpm, then filtered through filter paper and analyzed for the herbicide content. The concentration of both compounds in solutions was determined spectrophotometrically at λ = 283 nm (2,4-D) and λ = 278 nm (MCPA), using a Carry 3E spectrophotometer (Varian, Palo Alto, CA, USA). The calibration curves were obtained in the tested concentration range (0.05–1.0 mmol/L). Three repetitions were performed for each concentration value. With no outliers excluded, the calibration curves for both herbicides were linear with high *R*^2^ correlation coefficients (≥0.998). The equations for the regression line (*n* = 3) were *y* = 1.663*x* + 0.030 for 2,4-D and *y* = 1.396*x* + 0.039 for MCPA (where *y* is the absorbance and *x* is the concentration of the herbicide). To assess the quality of the methods, accuracy and precision expressed as relative standard deviation values (RSD) were determined. Accuracy at each concentration level was calculated by expressing the mean measured amount as a percentage of added amount. The accuracy for 2,4-D and MCPA was within 96.95–99.55% and 96.31–98.80%, respectively. The precision was for 2,4-D and MCPA within 1.88–4.29% and 1.93–4.62%, respectively.

The effect of solution pH on the adsorption of 2,4-D and MCPA was studied in the pH range from 2.5 to 10.0 for an initial herbicide concentration of 0.5 mmol/L. Appropriate pH values of the solutions were achieved by adding small amounts of 0.01 mol/L NaOH and/or 0.01 mol/L HCl. The adsorption kinetics of 2,4-D and MCPA from water on the carbon blacks was studied for the herbicide solutions with an initial concentration of 0.5 mmol/L. Equilibrium adsorption studies (adsorption isotherms) were carried out for six different initial concentrations of herbicides (0.2–1.0 mmol/L). The natural (original) pH of the herbicide solution (~4) was selected for the subsequent kinetic and equilibrium studies. The adsorption amount per mass of adsorbent was calculated from Equations (1) and (2):(1)$$q_{t}=\frac{\left(C_{0}-C_{t}\right)V}{m}$$
(2)$$q_{e}=\frac{\left(C_{0}-C_{e}\right)V}{m}$$
where: *q_e_* is adsorption capacity at equilibrium (mmol/g), *q_t_* is adsorption capacity after time *t* (mmol/g), *C*_0_ is initial concentration of the herbicide (mmol/L), *C_e_* is equilibrium herbicide concentration (mmol/L), *C_t_* is concentration after time *t* (mmol/L), *V* is the volume of the solution (L), and *m* is the mass of the adsorbent (g).

All adsorption experiments were performed in duplicate, and the average values were used for further calculations. The experimental error was about 5% (mean value).

## 3. Results and Discussion

### 3.1. Adsorbents Characterization

The low-temperature nitrogen adsorption–desorption isotherms as well as pore-size distribution curves for the carbon black samples are presented in Figure 1. 

The specific surface areas (*S*_BET_), as well as the pore volumes, were calculated from the N_2_ adsorption isotherms, and the results are listed in Table 2. The *S*_BET_ was calculated by the BET method, while the micro- and mesopores’ volumes were calculated using the t-plot method. It can be observed that the mesopore volumes are about 90% of the total porosity for all samples. As can be seen, the used modification methods of the carbon black surface caused a significant decrease in the parameter values describing its porous structure (*S*_BET_ as well as pore volumes). A greater reduction of these parameters can be observed for the carbon black with APTES deposited on its surface.

The *C* constant (the BET equation constant) is related to the energy of adsorption in the first adsorbed layer, and, consequently, its value is an indication of the magnitude of the adsorbent–adsorbate interactions. The value of the constant C indicates the type of adsorption isotherm in Brunauer’s classification; in our case, it will be type II isotherms [18].

The morphologies of carbon black samples visible in SEM images are shown in Figure 2. There, one can observe clear differences in the surface topography between the original carbon black and its modified samples. The oxidation of the surface increases its roughness and the deposition of APTES even more. 

The EDS analysis results of the carbon black samples are shown in Table 3. The oxidation of the original carbon black surface increases oxygen content by about 1.7 times. In the case of carbon black modified by APTES deposition, the oxygen content is slightly less than that for CB-NM. The Si (larger amount) and N observed in this sample are derived from APTES molecules. 

The results of the thermogravimetric analysis in the form of recorded TG curves for the carbon black samples used are shown in Figure 3. The proportion of weight loss in the temperature range of 180–700 °C is similar to the proportion of oxygen content in the surface layer of oxidized and original carbon black samples. This is due to the thermal decomposition of oxygen surface groups. In the case of the sample with deposited APTES, greater mass loss results from the boiling point of this compound being equal to 217 °C. 

The FTIR spectra of the carbon black samples are shown in Figure 4. The band around 3440 cm^−1^ corresponds to –OH stretching vibration from water adsorption or hydroxyl (phenolic) groups present in the carbon black surface. The absorption band near 1630 cm^−1^ is characteristic of C=O stretching vibration and can be attributed to polycyclic quinones. Both above bands are stronger for oxidized carbon black. For the carbon black sample with APTES present on its surface, the band corresponding to the asymmetric stretching of the Si–O–C bond and located at 1100 cm^−1^ is observed. The bending vibrations of NH_2_ amine near 1600 cm^−1^ are also observed.

### 3.2. Adsorption Study

#### 3.2.1. Effect of Solution pH

The physicochemical properties of the solution, including pH, significantly affect the adsorption of organic compounds by changing the electric charge on the adsorbent surface and the degree of ionization of the adsorbate. The effect of initial solution pH on the adsorption of 2,4-D and MCPA on carbon blacks is depicted in Figure 5.

The course of the *q_e_* = *f*(pH) dependence curves is very similar for both adsorbates, which is due to their similar physicochemical properties. The pK_a_ value for 2,4-D is 2.98, and for MCPA, it is 3.14 (Table 1). This means that in solutions with pH < pK_a_, these herbicides are present in a protonated form, while in solutions with pH > pK_a_ (above 2.98 and 3.14 for 2,4-D and MCPA, respectively), they appear in a dissociated form. The experimentally determined pH_PZC_ values for CB-NM, CB-Ox, and CB-APTES carbon blacks were 6.6, 6.1, and 7.2, respectively. The point of zero charge is the pH value at which the adsorbent surface has zero charge. In an environment with a pH above the point of zero charge, the carbon black surface is negatively charged, while when the pH is less than pH_PZC_, then the surface is positively charged. For 2,4-D and MCPA, the highest *q_e_* values were observed in an acidic environment with pH below their pK_a_ values. This suggests that the most favorable interaction occurs with electrically neutral adsorbate molecules and a positively charged carbon black surface. The worst adsorption occurred in an alkaline environment, as a result of the repulsive interaction of dissociated, negatively charged adsorbate molecules with a negatively charged adsorbent surface. In the case of unmodified and oxidized carbon black, a sharp decrease in adsorption was observed in the beginning pH range (2.5–6.0). A similar pH dependence of adsorption was observed for the removal of 2,4-D and/or MCPA from water by various activated carbons [16,19,20], carbon black [17], or lignite [21].

The same trend can be observed for the CB-APTES carbon black. A progressive decrease in adsorption efficiency with increasing solution pH is the result of interactions between the aminopropyl group of the adsorbent and the carboxylic group of herbicide molecules. Under acidic conditions, the adsorption is promoted by attractive electrostatic interaction between the protonated amine groups (–NH_3_^+^) on the adsorbent surface and negatively charged carboxylic groups of 2,4-D and MCPA (CB–NH_3_^+^······^−^OOC–R) [13,16]. APTES-modified carbon black, however, was characterized by greater resistance to pH changes, especially in the initial range, which is due to its alkaline nature and highest pH_PZC_ value.

#### 3.2.2. Adsorption Kinetics

Plots of *q_t_* = f(*t*) illustrating the adsorption kinetics of 2,4-D and MCPA on carbon blacks are shown in Figure 6.

The kinetic equilibrium was reached after about 60 min. The pseudo-first- (PFO) and pseudo-second-order (PSO) equations were used to describe the adsorption kinetics. The PFO kinetic model is expressed by the formula [22]:(3)$$\log\left(q_{e}-q_{t}\right)=\log q_{e}-\frac{k_{1}}{2.303}t$$
where *k*_1_ is the pseudo-first-order rate constant (1/min).

The PSO kinetic equation has the formula [22]:(4)$$\frac{t}{q_{t}}=\frac{1}{k_{2}q_{e}^{2}}+\frac{1}{q_{e}}t$$
where *k*_2_ is the pseudo-second-order rate constant (g/mmol∙min). 

To evaluate the best fit of the models used to the experimental data, the correlation coefficients (*R*^2^) and an error function, namely, normalized standard deviation (Δ*q*), were applied. This error function is expressed by Equation (5):(5)$$\Delta q=100\times \sqrt{\frac{\Sigma {\left[\left(q_{\mathrm{EXP}}-q_{\mathrm{CAL}}\right)/q_{\mathrm{EXP}}\right]}^{2}}{N-1}}$$
where *N* is the number of data points, while *q*_EXP_ and *q*_CAL_ are adsorption capacities that are experimental and obtained from the theoretical model.

A theoretical model with a higher correlation coefficient (*R*^2^) as well as a lower Δ*q* was found to fit the experimental data better.

The calculated values of correlation coefficients *R*^2^, normalized standard deviation (Δ*q*) values, rate constants (*k*_1_ and *k*_2_), and experimental adsorption capacities *q_e_* (*q*_EXP_), as well as those obtained from the kinetic equation (*q*_CAL_), are shown in Table 4. Looking at the value of the correlation coefficient and Δ*q* as the main criterion, it can be found that the adsorption kinetics of 2,4-D and MCPA on all three carbon blacks are described better by the PSO model (*R*^2^ ≥ 0.998) than the PFO model (*R*^2^ ≥ 0.988). This is also confirmed by the better agreement of *q*_EXP_ with the *q*_CAL_ values obtained for the PSO model. Thus, it can be concluded that the adsorption of 2,4-D and MCPA on carbon blacks followed the pseudo-second-order kinetic model.

The highest *q_e_* values were observed for APTES-modified carbon black, while the lowest values were obtained for oxidized carbon black (CB-Ox < CB-NM < CB-APTES). The adsorption rate constants *k*_2_ calculated from the PSO model increased in the order of CB-APTES < CB-Ox < CB-NM. Thus, both herbicides adsorbed fastest on unmodified carbon black and slowest on CB-APTES. Such a relationship appears to be closely correlated with the porous structure of the carbon blacks. The rate of adsorption is most influenced by the proportion of mesopores in the total porous structure of the adsorbent, as they constitute “transport pathways” by which adsorbate molecules are transported to the micropores. A high volume of micropores in practice usually means a high adsorption capacity of the adsorbate, while a high volume of mesopores generally translates into adsorption kinetics—the adsorption rate increases as the volume of mesopores increases. Thus, 2,4-D and MCPA were adsorbed fastest on CB-NM, which had the highest mesopore volume, and slowest on CB-APTES carbon black, which had the lowest mesopore volume.

Adsorption of a dissolved substance in solution is a multistep process and involves the mass transfer of the adsorbate (film diffusion), intraparticle diffusion, and localization of adsorbate molecules on the active sites of the adsorbent. The rate of the process is determined by those stages that occur most slowly, i.e., film diffusion or intraparticle diffusion (or both together) [23]. To understand which stage is the process that controls the rate of adsorption and, at the same time, to find out the mechanism of adsorption, kinetic models proposed by Weber–Morris and Boyd [22,23] were used.

The Weber–Morris model (intraparticle diffusion model) is expressed by the following equation:(6)$$q_{t}=k_{i}t^{0.5}+C_{i}$$
where *k*_i_ is the intraparticle diffusion rate constant (mmol/g min^−0.5^) and *C*_i_ is the thickness of the boundary layer.

According to this model, when the dependence of *q*_t_ vs. *t*^0.5^ is a straight line passing through the origin, then intraparticle diffusion is the most important process limiting the adsorption rate. On the other hand, when the graph does not pass through the origin and gives many linear segments, then adsorption kinetics can be controlled by film diffusion or both (film diffusion and intraparticle diffusion) simultaneously. Figure 7 shows the kinetic curves of the Weber–Morris equation. As can be seen, the plots are multilinear. The first, sharper stage is the boundary layer diffusion, while the second stage is the intraparticle diffusion process, where the intraparticle diffusion is the rate-limiting step. Moreover, the graphs do not pass through the origin (intercept ≠ 0). This indicates that there are many processes controlling adsorption and that intraparticle diffusion does not play a major role.

Boyd’s model is expressed by the following relationship:(7)$$\frac{q_{t}}{q_{e}}=1-\frac{6}{\pi^{2}}\overset{\infty }{\underset{1}{\sum }}\left(\frac{1}{n^{2}}\right)\exp\left(-n^{2}B_{T}\right)$$
where *B_T_* is a mathematical function of *q_t_*/*q_e_*.

This equation can be rearranged into the following forms:(8)$$B_{T}=\pi {\left(1-\sqrt{1-\frac{\pi }{3}\frac{q_{t}}{q_{e}}}\right)}^{2}$$
(9)$$B_{T}=-0.4977-\ln\left(1-\frac{q_{t}}{q_{e}}\right)$$

Equation (8) is used when $\frac{q_{t}}{q_{e}}<0.85$, while equation (9) is applied when $\frac{q_{t}}{q_{e}}>0.85$.

The kinetic model proposed by Boyd assumes that thin-liquid film on the adsorbent surface has the greatest effect on solute diffusion. By plotting *B_T_* versus *t*, it is possible to determine the rate-limiting step. Thus, if the plot *B_T_* = *f*(*t*) is nonlinear (or linear) and does not pass through the origin (intercept ≠ 0), then film diffusion will be the main determinant of adsorption, while if the plot *B_T_* = *f*(*t*) is a straight line and passes through the origin (intercept = 0), then adsorption is controlled by the intraparticle diffusion mechanism. Figure 8 shows the kinetic curves for the Boyd model, which are straight lines and do not pass through the origin of the coordinate system.

Based on Figure 7 and Figure 8, it can be concluded that the adsorption of 2,4-D and MCPA on carbon blacks is a complex process that is affected by both intraparticle diffusion and film diffusion, and where the film diffusion is the more important step in determining the rate of the whole adsorption process.

#### 3.2.3. Adsorption Isotherms

Isotherms of adsorption of both herbicides on the carbon blacks under equilibrium conditions are shown in Figure 9. For their mathematical description, the Langmuir, Freundlich, and Temkin isotherm equations [24] were used. The straight-line forms of these models can be expressed as follows:(10)$$\frac{C_{e}}{q_{e}}=\frac{1}{q_{m}}C_{e}+\frac{1}{q_{m}K_{L}}$$
(11)$$\ln q_{e}=\ln\;K_{F}+\frac{1}{n}\ln\;C_{e}$$
(12)$$q_{e}=\frac{RT}{b_{T}\;}\;\ln A_{T}\;+\frac{RT}{b_{T}\;}\;\ln C_{e}$$
where: *q_m_* (mmol/g) and *K_L_* (L/mmol) are the Langmuir isotherm parameters, *K_F_* ((mmol/g)(L/mmol)^1/n^) and *n* are the Freundlich isotherm constants, *b_T_* (J/mol) and *A_T_* (L/g) are the Temkin isotherm constants, *T* is the temperature (K), and *R* is the gas constant (8.314 J/mol·K).

The adsorption isotherm parameters were calculated from the slope and intercept of the linear plots of *C_e_*/*q_e_* vs. *C_e_*, ln*q_e_* vs. ln*C_e_*, and *q_e_* vs. ln*C_e_* for Langmuir, Freundlich, and Temkin isotherm models, respectively.

The Langmuir, Freundlich, and Temkin isotherm parameters are shown in Table 5. A high value of the correlation coefficient and low value of the normalized standard deviation indicate better agreement between experimental and predicted data using the Langmuir (*R*^2^ ≥ 0.996, Δ*q* ≤ 4.88%) equation than the Temkin (*R*^2^ ≥ 0.985, Δ*q* ≤ 12.69%) and Freundlich (*R*^2^ ≥ 0.951, Δ*q* ≤ 19.62%) isotherm models. The Langmuir isotherm assumes that there are no interactions between the adsorbate molecules, and that the monolayer adsorption occurs at homogeneous active sites in the adsorbent structure. Thus, the best correlation with the Langmuir isotherm model suggests monolayer adsorption of herbicides on homogeneous carbon black surfaces. Langmuir’s adsorption capacities were found to be 0.138, 0.340, and 0.124 mmol/g for adsorption of 2,4-D and 0.181, 0.348, and 0.139 mmol/g for adsorption of MCPA on CB-NM, CB-APTES, and CB-Ox, respectively. Both herbicides were most preferably adsorbed on APTES-modified carbon black and worst on oxidized carbon black (CB-APTES < CB-NM < CB-Ox). So, it is evident that the specific surface area was not the only factor determining the adsorption capacity. If this were the case, the adsorption capacity should be greatest for the adsorbent with the highest specific surface area (CB-NM, S_BET_ = 108 m^2^/g) and the worst for the one with the lowest BET surface area (CB-APTES, S_BET_ = 82 m^2^/g). Both herbicides adsorbed best on CB-APTES carbon black (with the lowest specific surface area). This shows that the surface chemistry of the adsorbent plays a more important role here than its porous structure. Zhu et al. [10] reported that nitrogen- and oxygen-containing functional groups can enhance the adsorption of 2,4-D. The results observed in this paper (CB-Ox < CB-NM < CB-APTES) are similar concerning the adsorbent with aminated surface and opposite to those obtained for adsorbents with an oxidized surface. The greater herbicide adsorption on the CB-APTES carbon black is due to the ionic interactions (electrostatic attraction) between protonated amino groups of the adsorbent and dissociated carboxyl groups of 2,4-D and/or MCPA molecules. This adsorption mechanism is favored especially at pH > pKa, when herbicides exist in dissociated form, and at pH < pH_PZC_, when the adsorbent surface is positively charged. In such conditions, the adsorption is promoted by attractive electrostatic interaction between the protonated amine groups (–NH_3_^+^) on the adsorbent surface and dissociated carboxylic groups of herbicides (CB–NH_3_^+^······^−^OOC–R) [13,16]. The pH_PZC_ concept seems to explain the adsorption behavior of 2,4-D and MCPA on APTES-modified carbon black well, suggesting that attractive and/or repulsive electrostatic interactions are one of the main mechanisms of phenoxyacetic herbicides adsorption on this adsorbent. These findings agree with the research results presented in other papers [10,11,12,13,14,15,16]. As in the paper [16], a significant reduction in adsorption on an adsorbent with an oxidized surface was observed here as well. This phenomenon can be attributed to the presence of surface-oxygen groups, which reduce the density of π-electrons in graphene layers and the hydrophobicity of the surface and consequently decrease the dispersive adsorption potential of the adsorbent [25]. Other reasons for this phenomenon may be the blockage of micropores and reduction of the surface area available to the adsorbate due to the hydration of polar oxygen groups, as well as the damage of thin pore walls, resulting in a decrease in the micropore volume [26].

Comparing the adsorption of the two herbicides, it can be seen that MCPA adsorbed better on each carbon black than did 2,4-D. Similar results have been reported for the adsorption of these herbicides on lignite [21] and various activated carbons [19,20,27,28]. The lower adsorption of 2,4-D compared to MCPA is probably due to the presence of two chlorine atoms on the aromatic ring. These substituents act as electron acceptor groups by weakening the dispersion interactions between the π electrons of the aromatic ring of 2,4-D and the π electrons of the graphene planes of the carbon adsorbents [19]. However, there are quite a few papers [29,30,31] that show the opposite phenomenon of better adsorption of 2,4-D than MCPA, suggesting that the preferable adsorption of 2,4-D or MCPA is determined by the individual properties of the adsorbent used in the study.

A comparison of 2,4-D and MCPA adsorption capacities on the carbon blacks used in this paper with other carbonaceous materials studied in other works is listed in Table 6.

## 4. Conclusions

Removal of 2,4-D and MCPA from aqueous solutions by adsorption using carbon blacks with different surface chemistry was demonstrated. Carbon black (CB-NM) samples were modified with hydrogen peroxide (CB-Ox) and aminopropyltriethoxysilane (CB-APTES). The adsorption of the herbicides was significantly influenced by pH. The kinetic experiments revealed that adsorption follows the pseudo-second-order kinetic model. The adsorption rate of 2,4-D and MCPA on all of the carbon blacks increased in the order of: CB-APTES < CB-Ox < CB-NM. The isotherm studies demonstrated that the Langmuir model provided the best fit for the adsorption of both herbicides on the carbon blacks. The adsorption efficiency of the 2,4-D and MCPA increased in the order of: CB-Ox < CB-NM < CB-APTES. Both the herbicides were preferably adsorbed on the APTES-modified carbon black due to the electrostatic attraction between protonated amino groups of the adsorbent and dissociated carboxyl groups of 2,4-D and/or MCPA molecules. The results showed that the surface chemistry of the adsorbent plays a more important role than its porous structure.

## Figures and Tables

**Figure 1 materials-15-08433-f001:**
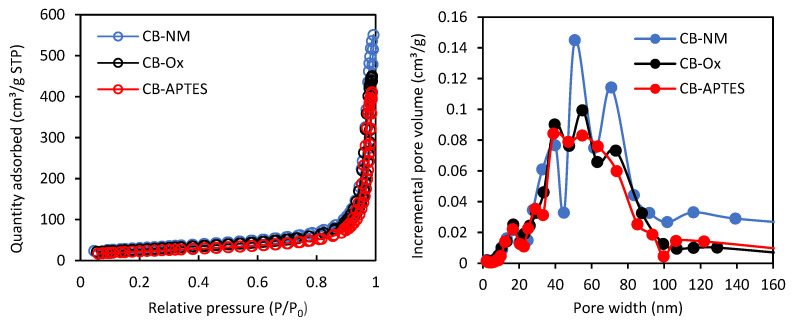
The nitrogen adsorption–desorption isotherms and pore size distribution curves for carbon black samples.

**Figure 2 materials-15-08433-f002:**
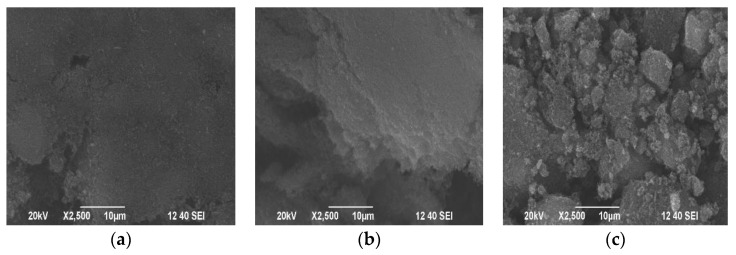
The SEM images of the carbon black samples: (**a**) CB-NM, (**b**) CB-Ox, and (**c**) CB-APTES.

**Figure 3 materials-15-08433-f003:**
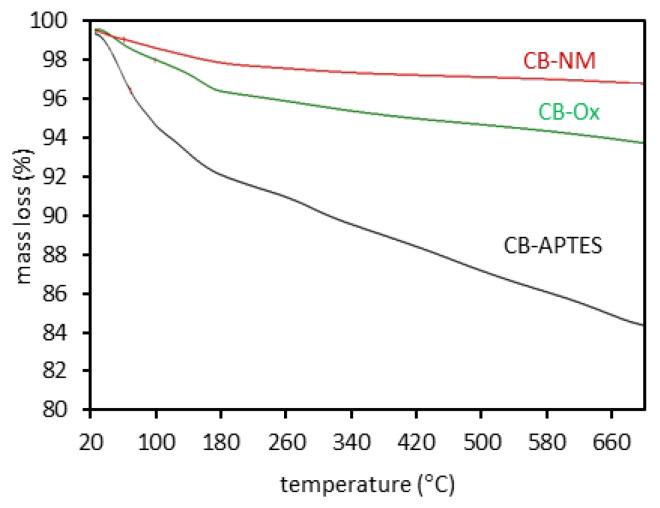
Mass loss (TG curves) for carbon black samples.

**Figure 4 materials-15-08433-f004:**
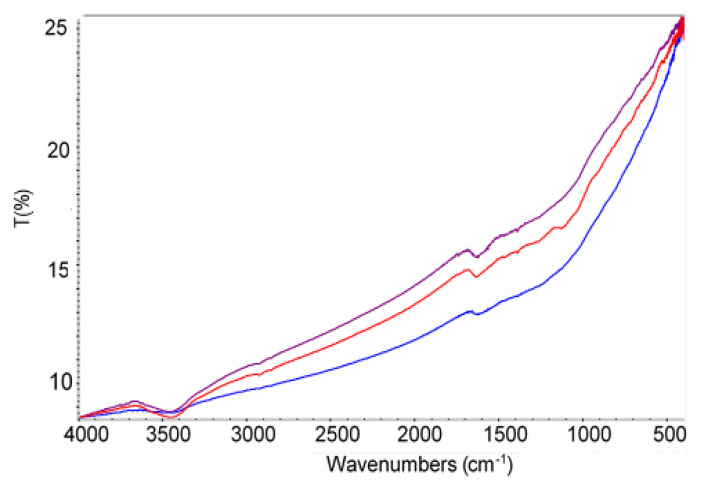
The FTIR spectra of the carbon black samples: CB-NM (blue line), CB-APTES (red line), and CB-Ox (purple line).

**Figure 5 materials-15-08433-f005:**
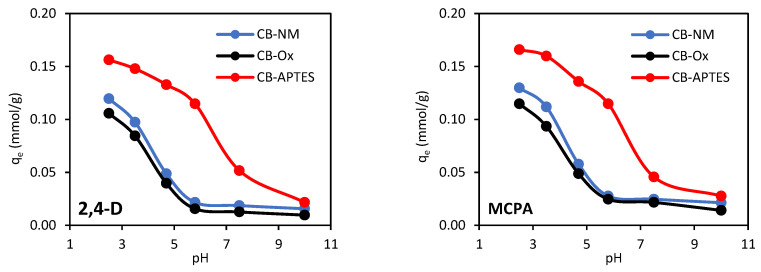
Effect of solution pH on the adsorption of the herbicides on carbon black samples.

**Figure 6 materials-15-08433-f006:**
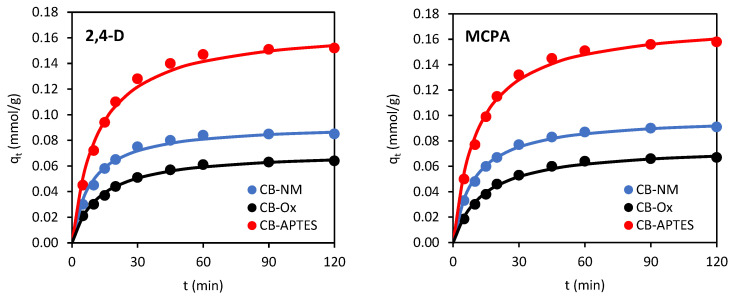
Adsorption kinetics of 2,4-D and MCPA on carbon black samples (line: fitting of PSO kinetic model).

**Figure 7 materials-15-08433-f007:**
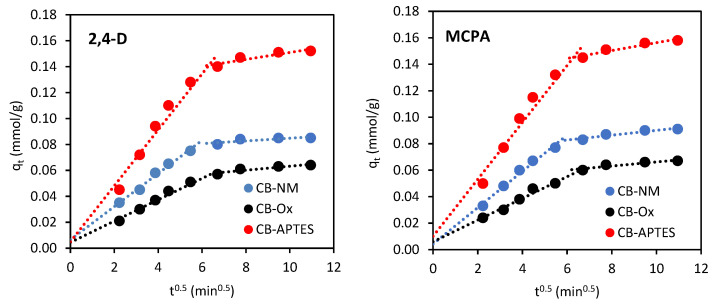
Weber–Morris diffusion model plots for adsorption of 2,4-D and MCPA on carbon black samples.

**Figure 8 materials-15-08433-f008:**
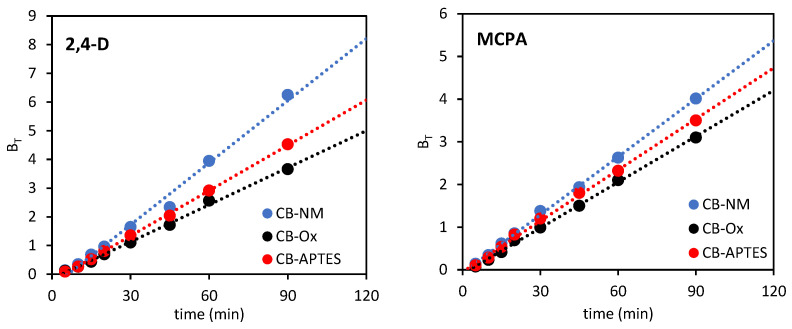
Boyd kinetic model plots for adsorption of 2,4-D and MCPA on carbon black samples.

**Figure 9 materials-15-08433-f009:**
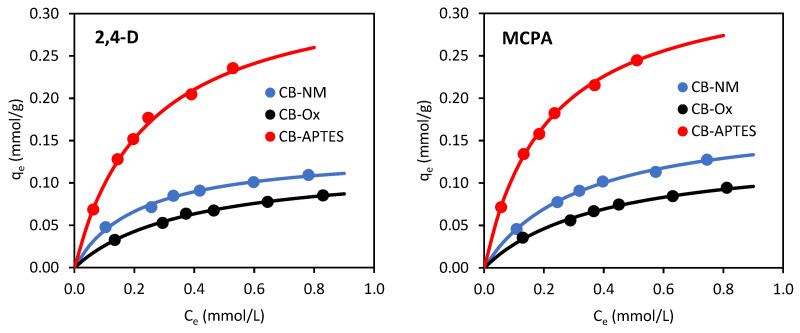
Adsorption isotherms of 2,4-D and MCPA from aqueous solutions on carbon black samples (line: fitting of Langmuir isotherm).

**Table 1 materials-15-08433-t001:** Physicochemical properties of the herbicides.

Herbicide	2,4-D	MCPA
CAS No.	94-75-7	94-74-6
Molecular formula	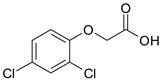	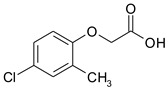
Molecular weight (g/mol)	221.04	200.62
Solubility in water (g/L)	0.89	0.82
pK_a_	2.98	3.14

**Table 2 materials-15-08433-t002:** Surface area and pore volumes of the carbon black samples.

Carbon Black	*S*_BET_ (m^2^/g)	*C* Constant	*V*_mi_ (cm^3^/g)	*V*_me_ (cm^3^/g)	*V*_t_ (cm^3^/g)
CB-NM	108	145	0.0503	0.0414	0.464
CB-Ox	95	155	0.0428	0.0398	0.441
CB-APTES	82	74	0.0372	0.0376	0.413

**Table 3 materials-15-08433-t003:** The EDS analysis results of the carbon black samples.

Carbon Black	C	O	Si	S	N
			(wt.%)		
CB-NM	95.6	4.1	-	0.3	-
CB-Ox	92.8	6.9	-	0.3	-
CB-APTES	94.6	3.7	1.1	0.2	0.4

**Table 4 materials-15-08433-t004:** Kinetic modeling data for adsorption of 2,4-D and MCPA on carbon black samples.

Adsorbate/Kinetic Model		Adsorbent	
**2,4-D**	CB-NM	CB-Ox	CB-APTES
*q*_EXP_ (mmol/g)	0.085	0.064	0.152
Pseudo-first-order			
*k*_1_ (1/min)	0.0734	0.0458	0.0551
*q*_eCAL_ (mmol/g)	0.087	0.054	0.137
*R* ^2^	0.990	0.975	0.988
Δ*q* (%)	8.91	15.93	11.12
Pseudo-second-order			
*k*_2_ (g/mmol∙min)	1.236	1.021	0.508
*q*_CAL_ (mmol/g)	0.092	0.071	0.169
*R* ^2^	0.998	0.998	0.999
Δ*q* (%)	3.53	3.71	2.86
**MCPA**	CB-NM	CB-Ox	CB-APTES
*q*_EXP_ (mmol/g)	0.091	0.067	0.158
Pseudo-first-order			
*k*_1_ (1/min)	0.0484	0.0469	0.0479
*q*_CAL_ (mmol/g)	0.071	0.059	0.127
*R* ^2^	0.982	0.990	0.985
Δ*q* (%)			
Pseudo-second-order			
*k*_2_ (g/mmol∙min)	1.0587	0.9471	0.5197
*q*_CAL_ (mmol/g)	0.098	0.074	0.170
*R* ^2^	0.999	0.999	0.998
Δ*q* (%)	1.92	2.33	4.01

**Table 5 materials-15-08433-t005:** The Freundlich, Langmuir, and Temkin isotherm constants for adsorption of 2,4-D and MCPA on carbon black samples.

Adsorbate/Isotherm		Adsorbent	
2,4-D	CB-NM	CB-Ox	CB-APTES
Freundlich			
*K*_F_ ((mmol/g)(L/mmol)^1/n^)	0.127	0.099	0.365
1/*n*	0.418	0.532	0.574
*R* ^2^	0.962	0.978	0.971
Δ*q* (%)	17.51	14.55	15.77
Langmuir			
*q*_m_ (mmol/g)	0.138	0.124	0.340
*K*_L_ (L/mmol)	4.621	2.622	4.061
*R* ^2^	0.996	0.997	0.996
Δ*q* (%)	4.32	3.87	4.88
Temkin			
*b*_T_ (kJ/mol)	43.03	22.34	37.07
*A*_T_ (L/g)	79.61	85.08	34.74
*R* ^2^	0.993	0.990	0.985
Δ*q* (%)	8.69	9.98	12.69
**MCPA**	CB-NM	CB-Ox	CB-APTES
Freundlich			
*K*_F_ ((mmol/g)(L/mmol)^1/n^)	0.157	0.110	0.383
1/*n*	0.531	0.534	0.556
*R* ^2^	0.977	0.989	0.951
Δ*q* (%)	13.84	10.01	19.62
Langmuir			
*q*_m_ (mmol/g)	0.181	0.139	0.348
*K*_L_ (L/mmol)	3.108	2.478	4.615
*R* ^2^	0.997	0.996	0.999
Δ*q* (%)	3.51	4.55	2.39
Temkin			
*b*_T_ (kJ/mol)	26.67	21.77	42.09
*A*_T_ (L/g)	58.67	76.65	31.42
*R* ^2^	0.985	0.991	0.991
Δ*q* (%)	11.99	9.69	9.87

**Table 6 materials-15-08433-t006:** Adsorption capacities of 2,4-D and MCPA on various carbonaceous materials.

Adsorbent	*S*_BET_ (m^2^/g)	Adsorption Capacity (q_m_), mmol/g		Ref.
		2,4D	MCPA	
CB-NM	108	0.138	0.181	this paper
CB-Ox	95	0.124	0.139	this paper
CB-APTES	82	0.340	0.348	this paper
Carbopack B carbon black	97	0.309	-	[17]
Vulcan XC 72 carbon black	227	0.325	-	[17]
SRB N762 carbon black	24.3	0.030	0.029	[32]
CA N660 carbon black	36.0	0.096	0.094	[32]
CO N539 carbon black	39.4	0.110	0.115	[32]
CA N375 carbon black	90.4	0.238	0.224	[32]
CO N375 carbon black	94.6	0.247	0.240	[32]
CA N115 carbon black	137	0.188	0.185	[32]
SRB N762 carbon black	24.3	0.030	0.029	[32]
carbon materials from cotton	27.4	0.149	-	[33]
carbon materials from filter paper	182	0.348	-	[33]
multiwalled carbon nanotubes	200	0.099	-	[34]
single-walled carbon nanotubes	597	2.001	-	[35]
reduced graphene oxide	512	1.222	-	[35]
biochar	523	0.039	-	[10]
K_2_CO_3_ activated biochar (KBC)	680	0.090	-	[10]
oxidated biochar (OKBC)	289	0.103	-	[10]
aminated biochar (NKBC)	691	0.058	-	[10]
aminosilane grafted carbons C_KIT-6_	834	0.497	-	[14]
Sorbo Norit activated carbon	1225	1.490	2.080	[19]
Ceca AC40 activated carbon	1201	1.560	2.599	[19]
Norit R3-ex activated carbon	1390	-	1.896	[16]
HNO_3_ treated R3-ex	1296	-	1.376	[16]
NH_3_ treated R3-ex	1212	-	2.703	[16]
activated carbon from willow	1280	2.310	2.413	[20]
activated carbon from miscanthus	1420	2.577	2.677	[20]
activated carbon from flax shives	1587	2.682	2.725	[20]
activated carbon from hemp shives	1324	2.446	2.460	[20]

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
