# Peer review of "Adsorption of 2,4-D and MCPA Herbicides on Carbon Black Modified with Hydrogen Peroxide and Aminopropyltriethoxysilane"

_materials, 2022, doi:10.3390/ma15238433_

Round 1

Reviewer 1 Report

Title : Adsorption of 2,4-D and MCPA herbicides on carbon blackmodified with hydrogen peroxide and 3 aminopropyltriethoxysilane

The authors present a study into the use of carbon blacks with different properties – origin carbon black (CB-NM) and carbon blacks modified with hydrogen per- 101 oxide (CB-Ox) and aminopropyltriethoxysilane (CB-APTES) on the adsorption of 2,4-D and MCPA.

The premise of the study and the idea of the prepared modified surface of carbon black is very interesting. Some specific questions/comments are below

  1. The line color in each graph for CB-NM, OX and APTES should be consistent to make it easier to read and compare
  2. Further explanation about surface modification between CB-Ox and CB-APTES  is needed. The FTIR almost show a similar result.
  3. The author should elaborate more on the detailed mechanism of 2,4-D and MCPA adsorption  onto modified carbon black.
  4. How much is the actual ratio/ percentage of the active functional group of modified carbon black  used for each 2,4-D and MCPA adsorption?
  5. How can you compare this system with already reported carbon black systems with similar surface modifications for adsorption?
  6. Discussion about adsorption efficiency compared to other adsorbents for herbicides previously investigated is required.

Reviewer 2 Report

Concerning the revised article (materials-2032973), I found that this paper has a good literature review, material and methods, and discussion in this interesting topic related to adsorption of 2, 4-D and MCPA herbicides on carbon black modified with hydrogen peroxide and aminopropyltriethoxysilane. Moreover, they have studied the adsorption processes in details such as kinetics, isotherm, and diffusion models. The results showed that the adsorption of the herbicides was pH-dependent and the observed in an acidic environment. The experimental data fit the pseudo-second-order kinetic model and mono-layer for adsorption isotherm by Langmuir model. The adsorption rate of both the herbicides increased in order: CB-APTES < CB-Ox < CB-NM 21 and was closely correlated with the mesoporous volume of the carbon blacks. Moreover, they showed that the surface chemistry of the adsorbent plays a more important role than its porous structure based on SEM images. Both herbicides were preferably adsorbed on APTES-modified carbon black and worst on oxidized carbon black (CB-APTES > CB-NM > CB-Ox). Based on these results, the authors have studied the adsorption process in a complete way and modeling all the obtained data by kinetic models and different isotherm models. For these reasons, I recommended to be publish in your journal.

Author Response

Dear Reviewer,

thank you for your very kind review.

Reviewer 3 Report

The manuscript "Adsorption of 2,4-D and MCPA herbicides on carbon black modified with hydrogen peroxide and aminopropyltriethoxysilane" has an important and actual subject of the research field.

The general presentation of this work is good and can be considered for publish.

However some little corrections are welcome:

- The SEM image can be presented at better resolution (magnitude and clarity)!

- The FTIR spectra of the carbon black samples are little suggestive, and if you have the possibility the RAMAN spectrum is recommended!

_ The pH control and monitoring must be a little more presented.

- The errors and validation of the analytical determination must give us!

Author Response

Please see the attanchment.

Reviewer 4 Report

Journal: Materials

Title: Adsorption of 2,4-D and MCPA herbicides on carbon black modified with hydrogen peroxide and aminopropyltriethoxysilane

Ms. ID.: materials-2032973

Legocka et al. present an experimental study in which carbon black N-220 was modified via oxidation by H2O2 and deposition of aminopropyltriethoxysilane. Starting material and the modified ones were characterized using SEM-EDX, FTIR spectroscopy, thermogravimetry, and N2 adsorption-desorption measurements. Then, the prepared carbon black materials were tested as adsorbents for the removal of 2,4-dichlorophe-15 noxyacetic acid and 2-methyl-4-chlorophenoxyacetic acid from aqueous solutions. The kinetics is analyzed in detail, using several most frequently used models for the adsorption of organic compounds of carbon-based materials, while the adsorption under equilibrium conditions was analyzed using Langmuir, Freundlich and Temkin isotherms. I find that the work is nicely organized and rather concise. Thus, I believe it is suitable for publication in Materials.

I have several suggestions for improvements:

- Fig. 1 – the symbols are so large that it is impossible to discriminate between the samples. However, as the manuscript could appear in color, the line is sufficient. It would be interesting to include the pore size distribution curves, along with adsorption isotherms.

- Table 2. – what is the C constant?

- SEM images show poor contrast, could it be improved?

- FTIR spectra are of poor quality, is there any way to improve them. Also, is it possible to obtain Raman spectra of these carbon samples? It would be very interesting to see Raman spectra, especially for the oxidized carbon sample.

- terminology could be better synchronized with the literature. For example, qe is usually called adsorption capacity

- please check Eqs. 1 and 2. Is V missing?

- Adsorption studies – please give your best to estimate uncertainties for data points and especially parameters obtained by fitting, like the rate constants, adsorption capacities (Table 4), and isotherm parameters presented in Table 5

- Weber-Morris model of intra-particle diffusion. Is it physically meaningful to have negative intercepts in this case? The discussion could be extended for this model; different linear parts in Fig. 7 could be ascribed to distinctive processes instead of saying there are many of them.

- What is the standard state for fitting the equilibrium adsorption data? The choice of the standard state affects all the constants appearing in the used isotherms.

- The used carbons have very modest specific surfaces. Would the conclusions be the same if one of them had an SBET of 2000 m2 g-1? In other words, there must be an interplay between textural properties and surface chemistry. Could Authors discuss this point?

Technical:

- In the submission system, the topic of this is the manuscript “Characterization of Electrochemical Materials”. This manuscript has nothing to do with it except the fact that carbon black is used in electrochemistry
